# Influences of Dietary Vitamin E, Selenium-Enriched Yeast, and Soy Isoflavone Supplementation on Growth Performance, Antioxidant Capacity, Carcass Traits, Meat Quality and Gut Microbiota in Finishing Pigs

**DOI:** 10.3390/antiox11081510

**Published:** 2022-08-01

**Authors:** Cui Zhu, Jingsen Yang, Xiaoyan Nie, Qiwen Wu, Li Wang, Zongyong Jiang

**Affiliations:** 1School of Life Science and Engineering, Foshan University, Foshan 528225, China; zhucui@fosu.edu.cn (C.Z.); 2111959058@stu.fosu.edu.cn (J.Y.); 20180410229@stu.fosu.edu.cn (X.N.); 2State Key Laboratory of Livestock and Poultry Breeding, Ministry of Agriculture Key Laboratory of Animal Nutrition and Feed Science in South China, Guangdong Provincial Key Laboratory of Animal Breeding and Nutrition, Maoming Branch, Guangdong Laboratory for Lingnan Modern Agriculture, Institute of Animal Science, Guangdong Academy of Agricultural Sciences, Guangzhou 510640, China; wuqiwen@gdaas.cn

**Keywords:** compound antioxidants, finishing pigs, carcass traits, gut microbiota, meat quality, vitamin E, selenium-enriched yeast, soy isoflavones

## Abstract

This study investigated the effects of dietary compound antioxidants on growth performance, antioxidant capacity, carcass traits, meat quality, and gut microbiota in finishing pigs. A total of 36 barrows were randomly assigned to 2 treatments with 6 replicates. The pigs were fed with a basal diet (control) or the basal diet supplemented with 200 mg/kg vitamin E, 0.3 mg/kg selenium-enriched yeast, and 20 mg/kg soy isoflavone. Dietary compound antioxidants decreased the average daily feed intake (ADFI) and feed to gain ratio (F/G) at d 14–28 in finishing pigs (*p* < 0.05). The plasma total protein, urea nitrogen, triglyceride, and malondialdehyde (MDA) concentrations were decreased while the plasma glutathione (GSH) to glutathione oxidized (GSSG) ratio (GSH/GSSG) was increased by compound antioxidants (*p* < 0.05). Dietary compound antioxidants increased loin area and b* value at 45 min, decreased backfat thickness at last rib, and drip loss at 48 h (*p* < 0.05). The relative abundance of colonic *Peptococcus* at the genus level was increased and ileal *Turicibacter_sp_H121* abundance at the species level was decreased by dietary compound antioxidants. Spearman analysis showed a significant negative correlation between the relative abundance of colonic *Peptococcus* and plasma MDA concentration and meat drip loss at 48 h. Collectively, dietary supplementation with compound antioxidants of vitamin E, selenium-enrich yeast, and soy isoflavone could improve feed efficiency and antioxidant capacity, and modify the backfat thickness and meat quality through modulation of the gut microbiota community.

## 1. Introduction

Pork represents one of the most consumed meats in China and many countries worldwide. The efficiency of pork production has improved tremendously over the past decades due to the great advances in animal breeding, nutrition, and management [1]. Meat quality is one of the most important factors that influence consumers’ preferences when purchasing pork [2]. With the increasing demand for high-quality pork, dietary manipulations have been widely applied for the improvement of meat quality and extension of shelf-life of pork products [3]. Among these, previous studies have demonstrated the positive effect of vitamin E in improving pork quality, oxidative stability, and shelf life [4,5]. Moreover, the dietary addition of selenium-enriched yeast at 0.3 mg/kg could significantly improve the antioxidant status and enhance the water-holding capacity in the muscle of finishing pigs [6]. The combination of vitamin E and selenium has been shown to effectively improve the performance of broilers [7], broiler breeders [8], and geese [9]. Previous experience also indicates that the dietary addition of soybean isoflavone at 20 mg/kg improved the antioxidant capacity and immune function in young piglets [10]. In addition, dietary supplementation with soy isoflavone could effectively modify the meat color and decrease the drip loss in growing-finishing pigs [11]. However, limited research could be found regarding the evaluation of the combination of vitamin E, selenium-enriched yeast, and soy isoflavones on meat quality in finishing pigs.

It is well-recognized that the composition and diversity of gut microbiota can be modulated by dietary nutrients, thus impacting the host’s health and diseases [12]. Notably, a recent study has indicated that the manipulation of gut microbiota could have the potential to improve meat quality and flavor by regulating the lipid metabolism of skeletal muscle in pigs [13]. However, the diet–microbiome–host interaction in finishing pigs remained largely unknown. Moreover, limited information is available concerning the effects of compound antioxidants on carcass traits and meat quality and their correlations with dynamic changes in gut microbiota when included in the diets of finishing pigs. We hypothesized that dietary compound antioxidants might affect the growth performance, antioxidant capacity, carcass traits, and meat quality through modulation of gut microbiota in finishing pigs. Therefore, the aim of this study was to evaluate the combination effects of vitamin E, selenium-enriched yeast, and soy isoflavone on the growth performance, antioxidant capacity, carcass traits, meat quality, and gut microbiota in finishing pigs, and to reveal the relationship between host responses to diets and gut microbiota after treatment with compound antioxidants.

## 2. Materials and Methods

### 2.1. Animals and Experimental Diets

The protocol for the animal trial was approved by the Animal Care and Use Committee of Foshan University, Guangdong, China. A total of 36 crossbred barrows (Duroc × Landrace × Yorkshire, DYL) [14] with similar initial body weight (BW) of 100.17 ± 1.11 kg were randomly assigned to control group or compound antioxidants group. The pigs in control group were fed the basal diets, while those in the compound antioxidants group were fed with the basal diets supplemented with 200 mg/kg vitamin E, 0.3 mg/kg selenium-enriched yeast, and 20 mg/kg soy isoflavone. Each treatment had 6 replicates (pens) of 3 pigs each. The vitamin E with a purity of 50% provided as DL-α-tocopherol acetate was purchased from the Jilin Beisha Pharmaceutical Co., Ltd. (Jilin, China). The selenium-enriched yeast, which contained 2500 mg/kg total Se content (98% purity in organic Se), was provided by Hubei Yaqi Biotechnology Co., Ltd. (Xiaogan, China). Moreover, the soy isoflavone, mainly consisting of daidzein (4, 7-dihydroxyisoflavone, 98% purity), was supplied by Guangdong Newland Feed Technology Co. Ltd. (Guangzhou, China). The doses of vitamin E, selenium-enriched yeast, and soy isoflavone were chosen according to previous reports [6,10,15,16,17,18].

The basal diets (Table 1) were formulated based on the nutritional requirements of National Research Council (NRC, 2012) [19] recommended for finishing pigs. The experiment lasted 28 days. Pigs were housed in slatted plastic flooring and had free access to feed and water throughout the whole experiment, and were deprived of feed overnight before the slaughter day. Each pig was weighed individually at the morning of d 1, 14, and 28 of the experiment. The growth performance, including average daily gain (ADG), average daily feed intake (ADFI), and feed to gain ratio (F/G) (or feed efficiency) during the experiment, was calculated accordingly on a pen basis (*n* = 6 of 3 pigs each).

### 2.2. Sample Collections

At the end of the experiment, six pigs per treatment (1 pig per replicate/pen) were selected according to the average body weight of the pen and subjected to fasting for 12 h. The blood samples were collected from overnight-fasting pigs by the inferior auricular vein punction. The plasma samples were then harvested by centrifugation at 1000× *g*, 4 °C for 10 min, and stored in 1.5 mL micro-tubes at −80 °C until use for analyzing biochemical parameters. After blood collection, the pigs were slaughtered after electrical stunning and dissect the carcass. Samples of the *longissimus lumborum* at the tenth rib of right carcass, and ileum and colon digesta were then collected, immediately frozen in liquid nitrogen followed by storage at −80 °C until analysis.

### 2.3. Analyses of Plasma Biochemical Parameters

The concentrations of plasma biochemical parameters including glucose, albumin, total protein, urea nitrogen, triglyceride, and total cholesterol were analyzed following the methods as described previously [20].

### 2.4. Analyses of Plasma Antioxidant Parameters and the mRNA Expression of Antioxidant Genes in Longissimus lumborum

The plasma contents of malondialdehyde (MDA), catalase (CAT), superoxide dismutase (SOD), glutathione peroxidase (GSH-Px), reactive oxygen species (ROS), and the ratio of glutathione (GSH) to glutathione oxidized (GSSG) (GSH/GSSG), were determined by the methods of respective commercial kits according to the manufacturer’s instructions (Nanjing Jiancheng Bioengineering Institute, Nanjing, China) as described previously [20].

The total RNA extraction of *longissimus lumborum* at tenth rib was extracted using Trizol reagent (Invitrogen, Carlsbad, CA, USA) according to the manufacturer’s protocols. After checking the purity and quality of extracted RNA, cDNA synthesis was performed using 1 µg total RNA with a PrimeScript™ II 1st Strand cDNA Synthesis Kit (Takara, Tokyo, Japan). The quantitative real-time PCR (qPCR) was performed in an ABI 7500 Mastercycler (Applied Biosystems, Waltham, MA, USA) with SYBR green PCR Mix (Takara, Dalian, China) according to the manufacturers’ instructions. The sequences of specific primers included *CAT*: 5′-TCCTGAGAGAGTTGTGCAT-3′(F) and 5′-CCAATTACCATCCTCTGTGT-3′(R), *SOD*: 5′-GGCCACATCAATCATAGCAT-3′(F) and 5′-TTAGAACAAGCGGCAATCTG-3′(R), *GSH-Px*: 5′-CCTCAAGTACGTCCGACCAG-3′(F) and 5′-TTCCATGCGATGTCATTGCG-3′(R), and *β-actin*: 5′-CATCGTCCACCGCAAAT-3′(F) and 5′-TGTCACCTTCACCGTTCC-3′(R). The relative mRNA abundance of the target genes was normalized, relative to β-actin as internal control, using the 2^−∆∆Ct^ method.

### 2.5. Carcass Traits

The hot carcasses were weighed individually after slaughter, and were processed into primal cuts. The dressing percentage was calculated accordingly. The content of abdominal fat at ventral midline was weighed after dissection from the left carcass according to the methods described previously [21]. The loin area was determined from *longissimus lumborum* muscle at the 10th and 11th rib of the left half-carcasses using the compensating planimeter (Model Q811; Xinanjiang Science Instrument Factory, Zhejiang, China). The backfat thickness was measured at first rib, 10th rib, last rib, and last lumbar with a vernier caliper (RH OMBI 5-32294, Guangzhou, China) on the right half-carcasses.

### 2.6. Meat Quality

The meat color was detected using *longissimus lumborum* muscle chops at 11th rib with Minolta chromameter (CR-410, Konica Minolta Sensing, USA). T, including L* (lightness), a* (redness), and b* (yellowness). The pH_45min_, pH_24h_, and pH_48h_ were assessed by a pH meter (HI 8242C, Beijing Hanna Instruments Science and Technology, Beijing, China). The drip loss was determined on *longissimus lumborum* muscle chops (2.5 cm thick) after trimming external fat and stored for 24 h and 48 h in a sealed plastic bag at 4 °C cold room. After storage, drip loss was recorded as the weight loss between the original weight of the chop and that at 24 and 48 h postmortem. The intramuscular fat (IMF) was analyzed for the ether extracts in the *longissimus lumborum* muscle by the methods of Association of Official Analytical Chemists (AOAC) [22]. Marbling score was graded using the color scoring cards (0–5 points) according to previous methods of National Pork Producers Council (NPPC) [23].

### 2.7. DNA Extraction, 16S rRNA Sequencing, and Bioinformatic Analysis

Microbial DNA was extracted from the ileal and colonic digesta samples by the method according to the manufacturer’s protocols of QIAamp DNA Stool Mini Kit (Qiagen, Hilden, Germany). The DNA concentrations were determined by NanoDrop 2000 spectrophotometer (Thermo Scientific, Wilmington, USA), and the DNA quality was checked by 1% agarose gel electrophoresis. The microbial sequencing was analyzed for V3 to V4 regions of the 16S rRNA gene with specific primers (341 F: 5′-CCTAYGGGRBGCASCAG-3′; 806 R: 5′-GGACTACNNGGGTATCTAAT-3′). The amplicon libraries were performed using the Ion Plus Fragment Library Kit (Thermo Fisher Scientific Inc., Waltham, MA, USA) and sequenced on an Illumina HiSeq250 platform (Novogene Technology Co., Ltd., Beijing, China). The sequencing data analysis was performed using QIIME2.0 software after removing the barcodes and primer sequences. The high-quality clean tags were clustered into operational taxonomic units (OTUs) with similarity more than 97%. The Venn diagram with shared and unique OTUs was used to identify the microbial similarities and differences between treatments. Taxonomic analysis was performed at the phylum and genus levels. The alpha-diversity (Shannon index and Chao 1 index) analyzed by Wilcox was conducted to study the complexity of species diversity using QIIME (V1.9.1). The principal coordinate analysis (PCoA) analysis based on binary-Jaccard distance, and Unweighted Pair-group Method with Arithmetic Means (UPGMA) clustering tree according to Bray–Curtis distance, showed the beta-diversity changes between treatments. The linear discriminant analysis effect size (LEfSe) analysis indicated the distinct microbial community variances between the two groups, and a further *t*-test was used to identify the differential bacteria at phylum, genus, and species levels.

### 2.8. Statistical Analysis

The pen was considered as the experimental unit for statistical analysis. The data were analyzed by Student’s *t*-test via SPSS software (version 22.0, IBM Corporation, Armonk, NY, USA). The results were expressed by the means ± standard error (SEM). The results showing values of *p* < 0.05 indicated a significant difference, while 0.05 < *p* < 0.10 indicated a trend. The correlations between variables and gut microbiota were performed by Spearman correlation analysis.

## 3. Results

### 3.1. Growth Performance

As shown in Table 2, dietary supplementation with compound antioxidants of 200 mg/kg vitamin E, 0.3 mg/kg selenium-enriched yeast, and 20 mg/kg soy isoflavone significantly decreased the ADFI and F/G at d 14–28 in finishing pigs (*p* < 0.05), but the ADFI and F/G during d 1–14 and d 1–28 were not significantly affected by dietary supplementation with the compound antioxidants (*p* > 0.05).

Moreover, there were no significant differences in BW at d 1, d 14, and d 28 as well as the ADG during d 1–14, d 14–28, and d 1–28 in finishing pigs between the two groups (*p* > 0.05) (Table 2).

### 3.2. Plasma Biochemical Indexes

Dietary supplementation with compound antioxidants of 200 mg/kg vitamin E, 0.3 mg/kg selenium-enriched yeast, and 20 mg/kg soy isoflavone for 4 weeks significantly decreased the plasma concentrations of total protein, urea nitrogen, and triglyceride (*p* < 0.05) (Table 3). However, there were no significant differences in plasma glucose, albumin, and cholesterol concentrations in finishing pigs between the two groups (*p* > 0.05) (Table 3).

### 3.3. Plasma Antioxidant Capacity and Antioxidant Gene Expression in the Muscle

As shown in Figure 1, dietary supplementation with compound antioxidants of 200 mg/kg vitamin E, 0.3 mg/kg selenium-enriched yeast, and 20 mg/kg soy isoflavones significantly decreased the plasma MDA concentration, and increased the plasma GSH/GSSG ratio (*p* < 0.05). Moreover, the *GSH-Px* mRNA expression in the *longissimus lumborum* muscle was significantly up-regulated by compound antioxidants (*p* < 0.05). However, there were no significant differences in CAT, T-SOD, GSH-Px, and ROS concentrations in the plasma nor in the mRNA expression of *CAT* and *SOD* genes in the *longissimus lumborum* muscle of finishing pigs (*p* > 0.05).

### 3.4. Carcass Traits

Dietary supplementation with compound antioxidants of 200 mg/kg vitamin E, 0.3 mg/kg selenium-enriched yeast, and 20 mg/kg soy isoflavones significantly increased the loin area, and decreased the backfat thickness at the last rib (*p* < 0.05) (Figure 2). However, there were no significant differences in carcass weight, dressing percentage, abdominal fat, and backfat thickness at the first rib, tenth rib, and last lumbar (*p* > 0.05).

### 3.5. Meat Quality

Dietary supplementation with compound antioxidants of 200 mg/kg vitamin E, 0.3 mg/kg selenium-enriched yeast, and 20 mg/kg soy isoflavones significantly increase the meat color b* value at 45 min, and decrease the drip loss at 48 h (*p* < 0.05) (Figure 3). There was a tendency of dietary compound antioxidants to slightly increase the muscle pH at 45 min in finishing pigs (*p* = 0.085). The L* value and a* value of meat color, intramuscular fat, and marbling score did not differ between the control and compound antioxidants groups (*p* > 0.05).

### 3.6. Analysis of Gut Microbiota Composition and Diversity

As shown in Figure 4, the common OTU in the ileum microbiota between the two groups is 133, with 84 and 93 unique OTUs in the control and compound antioxidants groups, respectively (Figure 4a). However, in the colon, there were 490 common OTUs, with 157 and 110 unique OTUs in the control and compound antioxidants groups, respectively (Figure 4b). The top 10 predominant phyla (Figure 4c) were Firmicutes, Proteobacteria, Tenercutes, unidentified Bacteria, Bacterodietes, Actinobacteria, Spiroachaetes, Cyanobacteria, Euyarchaecta, Melairnaeacteria, and others, which accounted for over 99% of the whole phyla in the ileum and colon. However, the dominant microbiota in ileal and colonic digesta at the genus level (top 10) included *Stenotrophomonas*, *Streptococcus*, unidentified *Clostridiales*, *Turicibacter*, *Romboutsia*, *Lactobacillus*, *Terrisporobacter*, *Sarcina*, *Halanaerobium*, and *Holdemanella* (Figure 4d).

For the microbial diversity shown in Figure 5, the rarefaction curve revealed that there was sufficient OTU coverage of bacterial composition in each group, with more coverage in the colon than ileum (Figure 5a). For the compound antioxidants group, the Shannon index of the ileal microbiota was different from that of the colonic microbiota (Figure 5b). Whereas the Chao 1 index of ileal microbiota was significantly different from that of colonic microbiota in the control group or compound antioxidants group (Figure 5c). However, there was no significant difference in the Shannon index or Chao 1 richness between the two dietary treatment groups. Furthermore, the PCoA based on the Bray–Curtis distance also revealed clear clustering of ileal or colonic digesta microbiota between the control and compound antioxidants groups (Figure 5d). Similar results were confirmed by the UPGMA analysis (Figure 5e).

The LEfSe analysis identified 34 and 6 biomarkers in the ileum (Figure 6a) and colon (Figure 6b), respectively. Notably, there were seven bacterial taxa enriched in the ileal digesta from the control group, including *Erysipelotrichales* (order), *Erysipelotrichaceae* (family), *Erysipelotrichia* (class), *Turicibacter* (genus), *Romboutsia* (genus), *Turicbacter_sp_H121* (species), and *Clostridium_butyricum* (species). Moreover, the ileal digesta of compound antioxidant groups enriched 27 bacterial taxa, which consisted of *Microbacteriaceae* (family), *Microbacterium_laevaniformans* (species), *Microbacterium* (genus), *Alphaproteobacteria* (class), *Pseudomonas_stutzeri* (species), *Pseudomonas* (genus), *Pseudomonadaceae* (family), *Bacteroidaceae* (family), *Bacteroides* (genus), *Alteromonadales* (order), *Alishewanella* (genus), *Alteromonadaceae* (family), *Rhizobiales* (order), Rhizobiaceze (family), unidentified_*Rhizobiales* (family), *Parococcus* (genus), *Rhodobacteraceae* (family), *Rhodobacterales* (order), *Caulobacterales* (order), *Pannonibacter_phragmitetus* (species), *Caulobacteraceae* (family), *Pannonibacter* (genus), *Helicobacter* (genus), *Helicobacteraceae* (family), *Helicobacter_pylori* (species), *Phyllobacterium* (genus), and *Campylobacterales* (order).

In the colonic digesta, however, only *Sarcina_ventricull* was enriched in the compound antioxidants group, while five bacterial taxa were found to be enriched in the control group, including unidentified *Bacteria* (class), unidentified Bacteria (phylum), *Halanaerobiales* (order), *Halanaerobium* (genus), and *Halanaerobiaceae* (family). Furthermore, LEfSe Cladogram (Figure 6c) also found that the enriched bacterial taxa in the colonic digesta included *Tannerallaceae* (family), *Christensenellaceae* (family), *Lachnospiraceae* (family), *Peptococcaceae* (family), *Ruminococcaceae* (family), and *Mollicutes* (order) by treatment with compound antioxidants, which mainly belonged to Firmicutes.

Furthermore, the microbial composition differences at different bacterial levels between intestinal segments or two groups were identified by Welch’s *t*-test (Figure 7). Firstly, at the phylum level, the relative abundance of Tenericutes in the colon was higher than that in the ileum of the control group. Moreover, the relative abundance of Bacteroidetes was significantly higher in the colon of the compound antioxidant group than that in the ileum (Figure 7a). Secondly, at the genus level, the relative *Turicibacter* in the ileum at the genus level was higher than that in the colon of the control group. Furthermore, the relative abundance of *Romboutsia* was lower, while those of unidentified_*Ruminococcaceae*, unidentified_*Lachnospiraceae*, and *Peptococcus* were higher in the colon when compared to the ileum of compound groups. In addition, an increase in the relative abundance of the genus *Peptococcus* was found in the colonic digesta of the compound antioxidants group than that in the control group (Figure 7b). Lastly, at the species level, a decrease in the relative abundance of *Turicibacter_sp_H121* was observed in the ileal digesta of the compound antioxidants group when compared to the control group (Figure 7c).

The Spearman correlation analysis (Figure 8) showed that there were significant correlations between gut microbiota in the top 35 genera and growth performance, antioxidant capacity, carcass trait, and meat quality in finishing pigs. The relative abundances of *Paracoccus* in the ileum (Figure 8a), as well as *Candidatus_Soleaferrea*, *Anaeroplasma*, *Butyricicoccus*, *Agathobacter*, *Halanaerobium*, and *Lactobacillus* in the colonic digesta (Figure 8b), were positively correlated with the ADFI at d 14–28 (*p* < 0.05). In addition, the F/G at d 14–28 was positively associated with the relative abundance of colonic *Porphyromonas*, and negatively associated with ileal *Microbacterium* and *Actinobacillus* as well as colonic *Marvinbryantia* (*p* < 0.05). Furthermore, the plasma MDA concentration was negatively correlated with the colonic *Peptococcus*, unidentified_*Lachnospiraceae*, *Subdoligranulum*, *Faecalibacterium*, and unidentified_*Ruminococcaceae* (*p* < 0.05). However, the b* value at 45 min was positively correlated with the relative abundances of ileal *Faecalibacterium* and *Microbacterium*, but was negatively correlated with ileal *Enterococcus* and colonic *Halanaerobium* (*p* < 0.05). The backfat thickness at the last rib was found to be positively associated with the relative abundance of *Terrisporobacter* in the ileal digesta (Figure 8a). Moreover, the drip loss at 48 h was negatively correlated with the relative abundances of *Paracoccus*, *Devosia*, *Microbacterium*, unidentified_*Cyanobacteria*, unidentified_*Corynebacteriaceae*, and *Streptococcus* in the ileal digesta (Figure 8a), as well as *Peptococcus*, unidentified_*Lachnospiraceae*, *Subdoligranulum*, *Blautia*, and *Holdemanella* in the colonic digesta (*p* < 0.05) (Figure 8b).

## 4. Discussion

In pig production, more and more attention has been attached to the importance of growth rate, carcass, and meat quality influencing consumer preferences, which could be improved by the manipulation of genetic selection, dietary interventions, and environmental management strategies. For dietary manipulation strategies, most studies have evaluated the effects of vitamin E, selenium-enriched yeast, or soy isoflavones on the performance and meat quality of animals, separately. However, limited information could be found regarding the combination of these compound antioxidants on growth performance, antioxidant capacity, carcass traits, meat quality, and gut microbiota in finishing pigs. In the present study, we found that dietary compound antioxidants with vitamin E, selenium-enriched yeast, and soy isoflavones had a lower ADFI and F/G, indicating an improvement in feed efficiency after dietary supplementation with compound antioxidants in finishing pigs. This was in accordance with a previous study whereby dietary vitamin E supplementation improved the growth performance, carcass yield percentage, and shelf life of goose [9]. Moreover, a previous study has also shown the effectiveness of selenium-yeast [24], polyphenols, and vitamin E [25], as well as soy isoflavones [26] in improving broiler growth performance, antioxidant capacity, and meat quality. In addition, a recent study has shown that dietary combination with vitamin E and Se supplementation significantly improved the growth performance and reduced the mortality of broilers under high ambient temperature [7]. Collectively, our results showed that compound antioxidants with vitamin E, selenium-enriched yeast, and soy isoflavones could effectively improve the feed efficiency of finishing pigs around 100 to 130 kg.

Blood biochemical indexes are often used to reflect the physiological and metabolic responses to diets in both animals and humans. For example, dietary selenium-enriched yeast supply at 0.3 mg/kg for 4 or 6 weeks could significantly lower the plasma concentrations of cholesterol and triglycerides in weanling pigs [27]. In addition, vitamin E treatment has been found to significantly decrease the serum urea nitrogen concentrations [28] and plasma triglyceride concentrations [29] in diabetic rats. Moreover, the plasma cholesterol concentrations were also significantly decreased by vitamin E combined with selenium treatments in hypercholesterolemic rabbits [30]. However, Lauridsen et al. found that plasma total cholesterol and triglyceride contents were not influenced by supplemental dietary vitamin E when included with rapeseed oils and copper in the diets of finishing pigs [31]. Notably, a systemic review by meta-analysis clearly demonstrated that isolated soy protein, isolated soy isoflavones, and soy protein containing soy isoflavones had a significant lowering effect on serum triglyceride and apolipoprotein B levels in postmenopausal women [32]. Consistently, plasma concentrations of total protein, urea nitrogen, and triglycerides were also decreased in finishing pigs when receiving compound antioxidants containing vitamin E, selenium-enriched yeasts, and soy isoflavones in the current study. Hence, these results may suggest an effective lowering effect on plasma lipids in finishing pigs after treatments with combinations of antioxidants.

Carcass characteristics are widely considered as economically important traits affecting swine production. The present study found that the loin area was increased while backfat thickness was decreased in finishing pigs of 100–130 kg by dietary compound antioxidants. Our results agreed with the previous study that dietary manipulation by antioxidants such as soy isoflavones effectively decreased fat and increased lean in growing barrows [11] due to its similarity to steroid estrogens in reducing body fat deposition. Furthermore, dietary vitamin E and Se supplementation significantly increased carcass composition by enhancing the weights of carcass, lean, and fat in broilers [7]. However, a previous study indicated that dietary long-term supplementation with selenium-enriched yeast during 25–105 kg did not affect the hot weight, backfat thickness, loin area, and loin percentage in growing–finishing pigs [33]. Accordingly, there were no significant alterations in the carcass weight and dressing percentage of finishing pigs treated with the combination of vitamin E, selenium-enrich yeast, and soy isoflavone for 4 weeks in the current study. This was consistent with a previous study that vitamin E supplementation did not affect the dressing percentage or fat thickness of finishing cattle [34]. Similarly, other studies also indicated that vitamin E supplementation did not affect the carcass yield measurements in growing–finishing pigs [35], or caused no significant differences in carcass composition in Berkshire- and Hampshire-sired pigs [36].

Meat quality often describes the attractiveness of meat to consumers. In recent years, the demand for high-quality pork has grown steadily with the continuously increasing living standards of consumers. Accumulative evidence has shown that excessive fat, poor color, and water-holding capacity account for the main quality concerns for consumer perceptions in the pork marketing chain [37,38,39]. Notably, dietary manipulation represents an important approach to meat quality and composition [40]. For example, dietary vitamin E supplementation significantly increased the L* value of breast meat color in finishing broilers [41] and tended to maintain meat redness (a*) of lambs [42]. However, a previous study in finishing pigs showed that vitamin E tended to increase b* values without affecting the pH, and L*, or a* values [36]. Consistently, we found that the b* value at 45 min postmortem was significantly increased, while drip loss at 48 h postmortem was decreased by dietary compound antioxidants with vitamin E, selenium-enriched yeast, and soy isoflavones applied in finishing pigs of 100–130 kg. Our results agreed with a previous study showing the improved overall preference of dry-cured ham by an antioxidant mixture containing vitamin E and polyphenols in the diets of finishing pigs at 95–130 kg [43]. Similarly, vitamin E supplementation could significantly reduce lipid oxidation and drip loss in Morkaraman male lambs [42] and improve pork shelf life in growing–finishing pigs [4]. Moreover, the drip loss was significantly decreased by dietary soy isoflavones addition in growing–finishing pigs [11]. Additionally, another study has demonstrated that dietary different levels of selenomethionine could significantly reduce the drip loss and cooking loss at 24 h postmortem in the *Longissimus lumborum* muscle of finishing pigs [44]. Importantly, a previous study also indicated that inorganic Se (sodium selenite) might have a detrimental effect on pork quality when compared to the organic Se (selenium-enriched yeast) [33]. Therefore, the improvement of meat quality could be partially explained by the use of organic Se supplementation as demonstrated in fattening pigs [16].

The meat quality (such as muscle pH, tenderness, drip loss, and meat color) was closely associated with its antioxidant capacity. Here, we found the plasma MDA was decreased and GSH/GSSG ratio was also increased by dietary compound antioxidant supplementation in finishing pigs. This could be due to the strong antioxidant properties of compound antioxidants with vitamin E, selenium-enriched yeast, and soy isoflavones. Indeed, selenium-enriched *Saccharomyces cerevisiae* could improve the meat quality by lowering drip loss and MDA contents as well as by enhancing the GSH-Px activity and total antioxidant capacity in Arbor Acres broiler chickens [45]. Similar results were found in finishing pigs whereby dietary 0.5 mg/kg Se increased serum GSH-Px activity and GSH concentration, and decreased MDA concentration [46]. One possible explanation for the decrease in MDA could be the protective effects of compound antioxidants on the reduction in lipid peroxidation production. Additionally, dietary supplementation with 200 mg/kg of vitamin E enhanced the blood GSH-Px activity in broiler chickens exposed to high temperature [25]. Due to the fact that Se is important for the regulation of GSH-Px activity, adding either selenium-enriched yeast (organic source) or sodium selenite (inorganic source) could improve serum GSH-Px activities in growing-fishing pigs at 30, 60, and 90 d [33]. However, the *GSH-Px* mRNA expression in the muscle of finishing pigs was significantly upregulated by compound antioxidants even though the activities of other antioxidant enzymes including GSH-Px, CAT, and SOD were not significant between the two groups in the current study. Thus, the combination of antioxidant levels applied here may not have been sufficient to bring about significant changes in these antioxidant enzyme activities in finishing pigs at 100–130 kg during the evaluated periods.

Gut microbiota plays important role in the regulation of host physiology, homeostasis, and health [47]. It is worth noting that the manipulation of gut microbiota has the potential to improve the meat quality and flavor of pigs through the gut microbiota–skeletal muscle axis [13]. A previous study has shown that transplantation of the gut microbiota induced an improvement of skeletal muscle mass in mice [47]. However, depletion of gut microbiota after treatment with antibiotics for 21 days would impair the optimal muscle function in mice [48]. Studies in mice have shown that consumption of different doses of vitamin E could affect the gut microbiota composition, especially for the ratio of Firmicutes to Bacteroidetes [49]. Another study in Tibetan sheep also found selenium yeast affected rumen bacterial communities and microbial fermentation, with a medium and high level of selenium yeast suitable for the survival of Bacteroidetes and Firmicutes, respectively [50]. Notably, soy isoflavones significantly modulated the composition of gut microbiota to alleviate depression in rats [51]. Hence, we hypothesized that the attenuation of performance, antioxidant capacity, carcass traits, and meat quality may be closely associated with the modulation of gut microbiota. Therefore, the composition and diversity of the gut microbiota were assessed by high-throughput sequencing of 16S rRNA gene amplicons. Consistent with previous studies, Firmicutes and Bacteroidetes were the dominant phyla in finishing pigs. PCoA analysis also showed different microbial community structures and compositions between two dietary treatment groups. LEfSe analyses showed that the compound antioxidant group was dominated by potential biomarkers with classified strains of bacteria belonging to the phylum Firmicutes in finishing pigs. Moreover, dietary compound antioxidants increased the relative abundance of colonic *Peptococcus* at the genus level and decreased ileal *Turicibacter_sp_H121* abundance at the species level. A Previous study showed a positive correlation between the genera *Turicibacter* and *Romboutsia* and serum high-density lipoprotein cholesterol and SOD levels in rats fed by high-fat diet [52]. Moreover, the intestinal microbiota of genera *Peptococcus* and *Eubacterium* exhibited strong positive correlations with growth traits including BW and ADG in growing pigs [53]. Consistently, in the present study, the Spearman correlation analysis also found that the plasma MDA concentration and drip loss were negatively correlated with the relative abundance of colonic *Peptococcus*. This suggested that the increased genera *Peptococcus* abundance in the colonic digesta was beneficial for decreasing the plasma MDA and drip loss of meat in finishing pigs. Interestingly, a previous study has also found that *Peptococcus* and *Lactococcus* were significantly enriched in the colon of finishing Landrace pigs with high or low feed conversion ratio (FCR), suggesting the possibility of targeting alteration of intestinal microbial communities for improved performance [54]. Furthermore, the relative abundance of *Lactobacillus* in the colonic digesta was positively correlated with the ADFI of finishing pigs. This was consistent with the previous report that cecal *Lactobacillus* positively correlated with broiler growth performance [55]. Thus, the dynamic changes in the gut microbiota community would partially explain the potential improvements in growth performance, antioxidant capacity, carcass traits, and meat quality found in the current study. However, further studies by techniques such as fecal microbiota transplantation would be needed to elucidate the interactions between these antioxidant compounds with dynamic changes in gut microbiota as well as the role and potential mechanism of gut microbiota in enhanced antioxidant capacity and meat quality by dietary compound antioxidants.

## 5. Conclusions

Taken together, these results suggest that dietary supplementation with 200 mg/kg vitamin E, 0.3 mg/kg selenium-enriched yeast, and 20 mg/kg soy isoflavone could significantly increase the feed efficiency during the last two weeks before slaughter, improve the antioxidant capacity and meat color, and reduce backfat thickness and drip loss in finishing pigs during 100 to 130 kg. These alterations of host responses to dietary compound antioxidants might be closely associated with the changes in ileal and colonic microbiota composition in finishing pigs. This study might provide novel insights into future applications of compound antioxidants for the livestock industry by focusing on the importance of diet–microbiome–host interactions. However, some limitations of this study should be noted. Further investigations concerning manipulating dietary antioxidant mixtures in finishing pigs are necessary to be conducted on a larger scale for confirming their effective combination on meat quality and antioxidant function through targeting the gut microbiota–muscle axis.

## Figures and Tables

**Figure 1 antioxidants-11-01510-f001:**
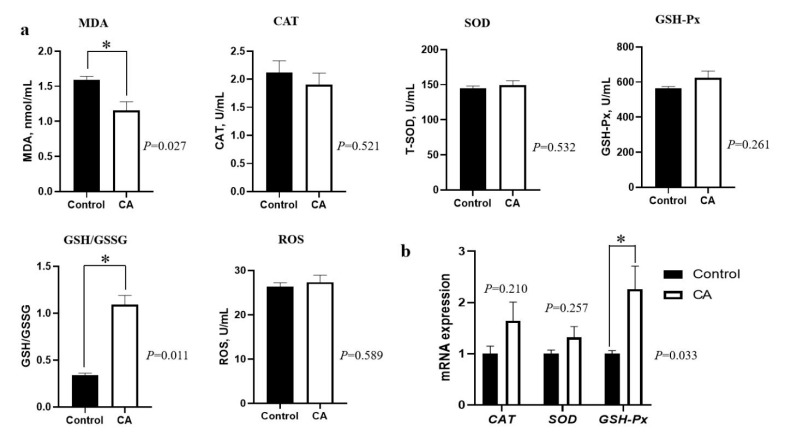
Effects of dietary supplementation with compound antioxidants on plasma antioxidant capacity and the mRNA expression of antioxidant genes in the *longissimus lumborum* muscle of finishing pigs. Abbreviations: CA, compound antioxidants of 200 mg/kg vitamin E, 0.3 mg/kg selenium-enriched yeast, and 20 mg/kg soy isoflavone; MDA, malondialdehyde; CAT, catalase; SOD, superoxide dismutase; GSH-Px, glutathione peroxidase; GSH/GSSG, the ratio of glutathione (GSH) to glutathione oxidized (GSSG); ROS, reactive oxygen species. * indicates significant difference between two groups (*p* < 0.05).

**Figure 2 antioxidants-11-01510-f002:**
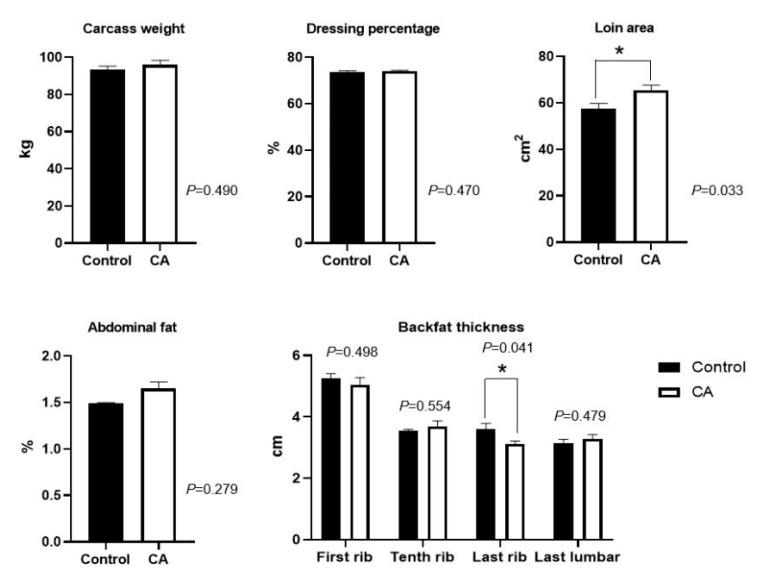
Effects of dietary supplementation with compound antioxidants on carcass traits in finishing pigs. Abbreviations: CA, compound antioxidants of 200 mg/kg vitamin E, 0.3 mg/kg selenium-enriched yeast, and 20 mg/kg soy isoflavone; * indicates significant difference between two groups (*p* < 0.05).

**Figure 3 antioxidants-11-01510-f003:**
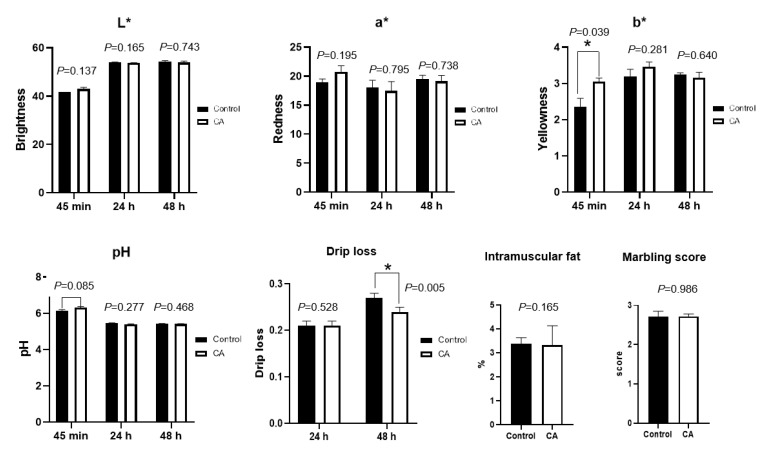
Effects of dietary supplementation with compound antioxidants on meat quality in finishing pigs. Abbreviations: CA, compound antioxidants of 200 mg/kg vitamin E, 0.3 mg/kg selenium-enriched yeast, and 20 mg/kg soy isoflavone; L* indicates lightness, a* represents for redness, and b* stands for yellowness. * Indicates significant difference between two groups (*p* < 0.05).

**Figure 4 antioxidants-11-01510-f004:**
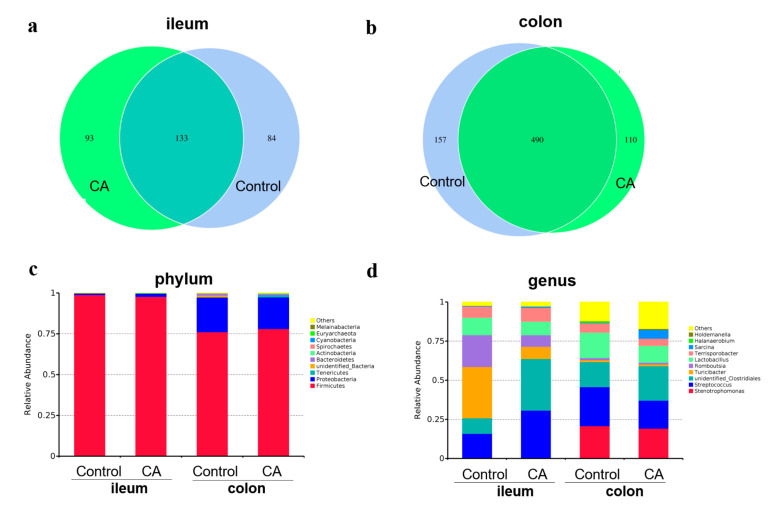
Dietary supplementation with compound antioxidants on the Venn diagram and relative abundance of gut microbiota in finishing pigs. Abbreviations: CA, compound antioxidants of 200 mg/kg vitamin E, 0.3 mg/kg selenium-enriched yeast, and 20 mg/kg soy isoflavone. (**a**) The Venn diagram of ileal microbiota. (**b**) The Venn diagram of colonic microbiota. (**c**) The gut microbiota structure at phylum level. (**d**) The gut microbiota structure at genus level.

**Figure 5 antioxidants-11-01510-f005:**
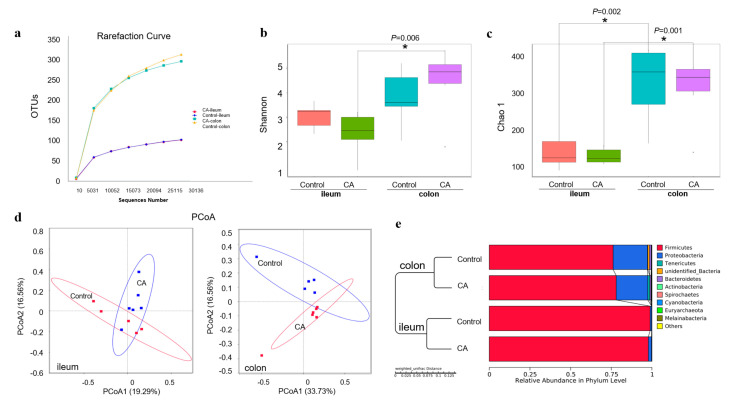
Dietary supplementation with compound antioxidants on the diversity of gut microbiota in finishing pigs: (**a**) Rarefaction curve. (**b**) Shannon index and (**c**) Chao 1 richness were analyzed by Wilcox. (**d**) The PCoA analysis based on binary Jaccard. (**e**) UPGMA clustering based on Bray–Curtis. Abbreviations: CA, compound antioxidants of 200 mg/kg vitamin E, 0.3 mg/kg selenium-enriched yeast, and 20 mg/kg isoflavone. * Indicates significant difference between ileum and colon (*p* < 0.05).

**Figure 6 antioxidants-11-01510-f006:**
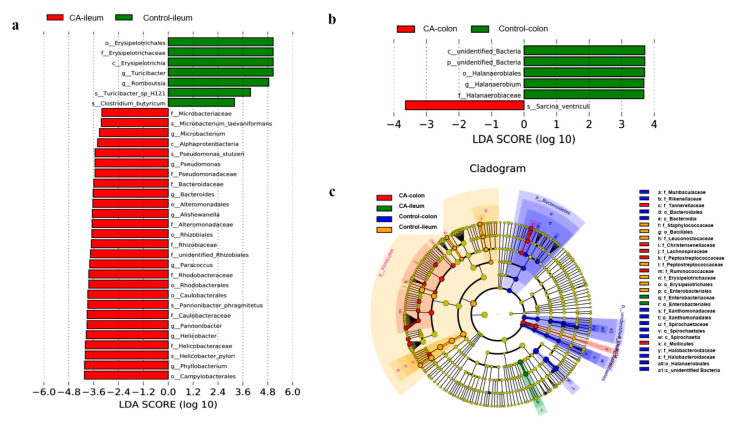
Linear discriminant analysis (LDA) effect size (LEfSe) analysis (*p* < 0.05, LDA > 3.0) of gut microbiota in finishing pigs: (**a**) The LEfSe analysis of ileal microbiota. (**b**) The LEfSe analysis of colonic microbiota. (**c**) The LEfSe cladogram in both ileal and colonic microbiota. Abbreviations: CA, compound antioxidants of 200 mg/kg vitamin E, 0.3 mg/kg selenium-enriched yeast, and 20 mg/kg soy isoflavone. The prefix “p” represented for phylum, while “c”, “o”, “f”, “g”, and “s” stand for class, order, family, genus, and species, respectively.

**Figure 7 antioxidants-11-01510-f007:**
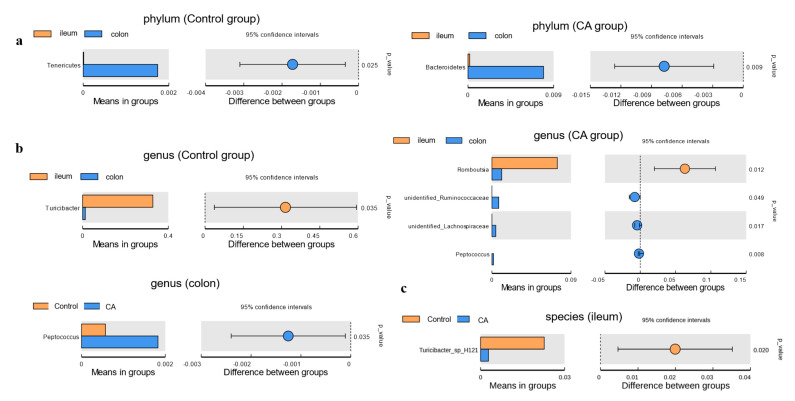
*T*–test analysis for the significant changes in differential gut microbiota at different levels: (**a**) Phylum. (**b**) Genus. (**c**) Species. Abbreviations: CA indicates compound antioxidants of 200 mg/kg vitamin E, 0.3 mg/kg selenium-enriched yeast, and 20 mg/kg soy isoflavone.

**Figure 8 antioxidants-11-01510-f008:**
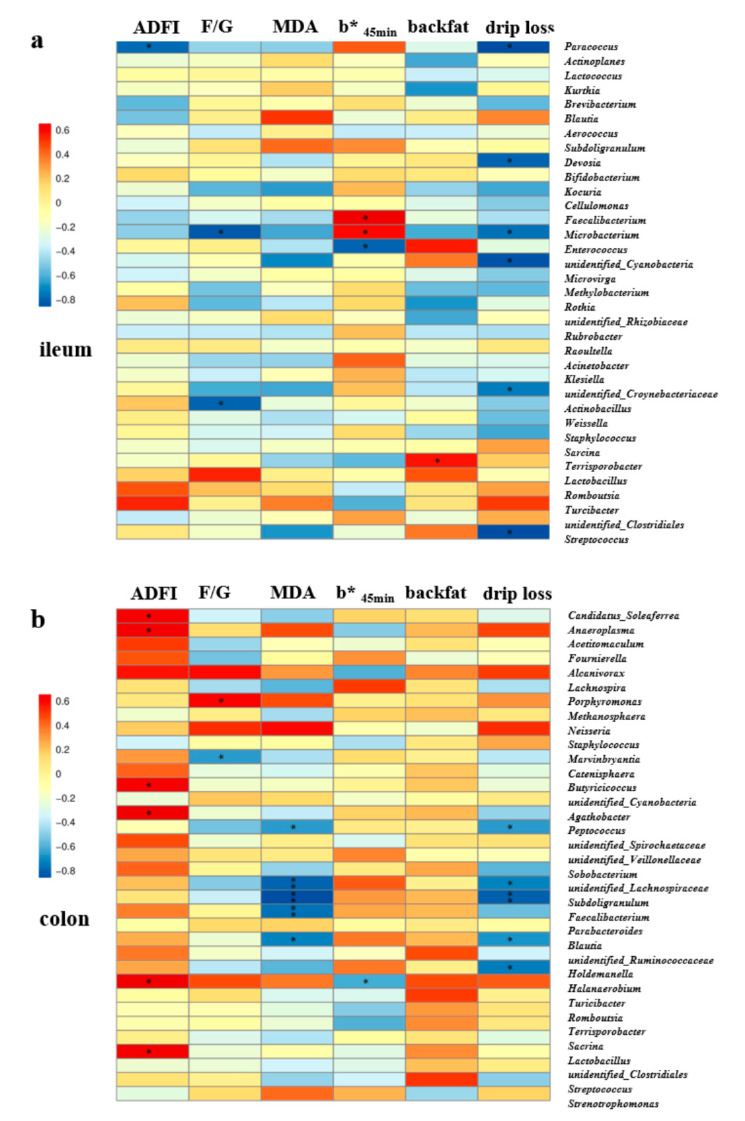
The Spearman correlation analysis of gut microbial composition with growth performance, antioxidant capacity, carcass traits, and meat quality in finishing pigs. Spearman correlation coefficients of ADFI and F/G at d 14–28, plasma MDA concentration, b* value at 45 min, backfat thickness at last rib, and drip loss at 48 h with ileal (**a**) and colonic (**b**) microbiota at genus level, are displayed. The heatmap with red indicated a positive correlation, while blue represented a negative correlation. Abbreviations: ADFI, Average daily feed intake; F/G, Feed to gain ratio. MDA, malondialdehyde. * *p* < 0.05 and ** *p* < 0.01.

**Table 1 antioxidants-11-01510-t001:** Ingredient and nutrient composition of the basal diet.

Ingredient	%	Nutrient Composition ^1^	
Corn	85.6	DE, MJ/kg	14.43
Soybean meal	9.73	ME, MJ/kg	14.07
Soybean oil	1.00	NE, MJ/kg	10.24
L-Lysine HCl	0.46	CP, %	12.01
DL-Methionine	0.05	Ca, %	0.51
L-Tryptophan	0.06	Total phosphorus, %	0.47
L-Threonine	0.15	Available phosphorus, %	0.28
Limestone	0.65	SID ^3^ Lysine, %	0.83
CaHPO_4_	1.00	SID Methionine, %	0.25
NaCl	0.30	SID Methionine + Cystine, %	0.47
Premix ^2^	1.00	SID Threonine, %	0.53
Total	100.00	SID Tryptophan, %	0.17

^1^ Calculated values. ^2^ Supplied per kg of diet: vitamin A, 1300 IU; vitamin D_3_, 150 IU; vitamin E, 11 mg; vitamin K_3_, 0.50 mg; vitamin K, 0.50 mg; biotin, 0.05 mg; choline, 0.30 mg; folic acid, 30 mg; calcium pantothenate, 7 mg; vitamin B_1_, 1 mg; vitamin B_2_, 2 mg; vitamin B_6_, 5 mg; vitamin B_12_, 0.05 mg; Cu, 3 mg; Fe, 40 mg; Zn, 50 mg; Mn, 2 mg; I, 0.14 mg; Se, 0.15 mg. ^3^ SID, standard ileal digestibility.

**Table 2 antioxidants-11-01510-t002:** Effects of dietary supplementation with compound antioxidants on growth performance in finishing pigs.

Item	Control Group	CA Group	*p* Value
Body weight (BW), kg			
d 1	100.42 ± 1.84	99.92 ± 1.57	0.841
d 14	113.67 ± 1.99	113.32 ± 2.09	0.907
d 28	128.30 ± 2.83	130.17 ± 1.88	0.851
Average daily feed intake (ADG), kg/d			
d 1–14	0.95 ± 0.08	0.96 ± 0.14	0.942
d 14–28	0.88 ± 0.08	0.96 ± 0.10	0.596
d 1–28	0.92 ± 0.04	0.96 ± 0.08	0.614
Average daily feed intake (ADFI), kg/d			
d 1–14	3.39 ± 0.05	3.51 ± 0.32	0.709
d 14–28	3.01 ± 0.17	2.63 ± 0.12 *	0.030
d 1–28	3.20 ± 0.07	3.06 ± 0.18	0.496
Feed to gain ratio (F/G)			
d 1–14	3.69 ± 0.29	3.79 ± 0.19	0.799
d 14–28	3.54 ± 0.37	2.88 ± 0.26 *	0.042
d 1–28	3.55 ± 0.22	3.25 ± 0.17	0.226

Abbreviations: CA, compound antioxidants of 200 mg/kg vitamin E, 0.3 mg/kg selenium-enriched yeast, and 20 mg/kg soy isoflavone. * Indicates significant difference within the same line between two groups (*p* < 0.05).

**Table 3 antioxidants-11-01510-t003:** Effects of dietary supplementation with compound antioxidants on plasma biochemical indexes in finishing pigs.

Item	Control Group	CA Group	*p* Value
Glucose, mmol/L	4.33 ± 0.27	4.63 ± 0.27	0.441
Albumin, g/L	39.70 ± 1.30	41.87 ± 0.80	0.211
Total protein, μg/mL	69,527.88 ± 1577.40	63,205.05 ± 934.74 *	0.009
Urea nitrogen, ng/mL	2.82 ± 0.28	2.14 ± 0.08 *	0.032
Triglyceride, mmol/L	0.27 ± 0.04	0.17 ± 0.01 *	0.001
Total cholesterol, mmol/L	3.85 ± 0.22	4.05 ± 0.27	0.590

Abbreviations: CA, compound antioxidants of 200 mg/kg vitamin E, 0.3 mg/kg selenium-enriched yeast, and 20 mg/kg soy isoflavone; * indicates significant difference within the same line between two groups (*p* < 0.05).

## Data Availability

Data are contained within the article except that the original contributions of 16S RNA sequencing data presented in the study are publicly available. These data can be found here: Submission ID: SUB11504085, BioProject ID: PRJNA841049. The direct link/URL to these data is https://submit.ncbi.nlm.nih.gov/subs/sra/SUB11504085/overview (accessed on 20 June 2022).

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
