# Peer review of "Influences of Dietary Vitamin E, Selenium-Enriched Yeast, and Soy Isoflavone Supplementation on Growth Performance, Antioxidant Capacity, Carcass Traits, Meat Quality and Gut Microbiota in Finishing Pigs"

_antioxidants, 2022, doi:10.3390/antiox11081510_

Round 1
Reviewer 1 Report
This study provides some interesting information, but the following points should be addressed before publication.
1. The words “ADFI” and “F/G” are not widely used. Therefore, it is necessary to give their full names in the first description of Abstract (Line 5).
2. The rationale about the dose of 67 mg/kg vitamin E, 0.3 mg/kg selenium-enriched yeast, and 20 mg/kg soy isoflavones in the diet should be explained.
3. There is no information of the status of oxidative stress in the muscle.
4. Is there significant relationship of the microbiota with meat quality ?
Already, similar studies have been reported. Therefore, authors should clearly explain what is the difference of this study with the previous studies reported and the impact of this study.
Reviewer 2 Report
The aim of this study was to evaluate the combination effects of vitamin E, selenium-enriched yeast, and soy isoflavones on the growth performance, antioxidant capacity, carcass traits, meat quality,and gut microbiota in finishing pigs, and the relationship with gut microbiota.
The study of the effects of antioxidants on the antioxidant capacity of the animal and the quality of meat is certainly not new and an extensive bibliography is available on the subject. However, the interrelations with the intestinal microbiota take on an original character.
The presentation of all the results in graphical form makes it not easy to evaluate the reported data, moreover in this way only the significance and not the p values are reported which can be very useful.
Line 68 ‘0.3 mg/kg selenium-enriched yeast’ I think it means 0.3 mg of Se apported by selenium-enriched yeast. Therefore it is useful to report the quantity of added yeasts and the% of Se.
Line 70. ‘soy isoflavones’ contain different isoforms in four chemical forms: aglycone (daidzein, genistein and glycitein), glucoside (daidzin, genistin and glycitin), acetylglucoside (acetyldaidzin, acetylgenistin and acetylglycitin) and malonylglucoside (malonyldaidzin, malonylgenistin and malonylglycitin). It is necessary to better specify the composition of the product used.
Conclusion
not all conclusions are supported by the results reported: in particular the FE did not improve significantly in the period 1-28. Some caution is also needed because they are the results of performances on 18 animals
Reviewer 3 Report
Dear authors,
The work entitled 'Dietary vitamin E, selenium-enriched yeast, and soy isofla- 2 vones improve the antioxidant capacity and meat quality by 3 modulation of gut Mmicrobiota in finishing pigs´ has been submitted for review. The document is important to animals science and it contributes to know the use of new foods in feeding pig.
However, I consider that the document is not correctly focused since the conclusions are not correctly supported by the experimental approach. Knowing the initial state of the microbiota of the study animals is required to know the final state of the study microbiota.
On the other hand, the study of the influence of the proposed diet based on the use of antioxidants should include an analysis of the fatty acid profile. In the work mention is made of the importance of the quality of pork. The fatty acid profile is important in the quality of the meat and its nutritional value.
The authors should consider this aspect for a new submission of the document.
Some sugestions are reported following:
Title:
The title should be abbreviated and not be shown as a partial conclusion.
Abstract
Abbreviations in the abstract don´t have been supported.
Introduction
Referencies to combinated use of 200 mg/kg vitamin E, 0.3 190 mg/kg selenium-enriched yeast, and 20 mg/kg soy isoflavones combined in animals feed could be reported.
Material and methods
Table 1: Is the last column unit correct? %?
The authors must justify the reason for choosing the diet 200 mg/kg vitamin E, 0.3 190
mg/kg selenium-enriched yeast, and 20 mg/kg soy isoflavones
References to the animals origin are necessaries. Around weight for 100kg is not a good information. What was the initial fatness of the animals?
Lines 101-105: Is there evidence that they are tested methods? Put validation reference.
Lines 124-125: “The content of abdominal fat was weighed after dissection from the carcass”. This method need a reference. What is the abdominal fat? What do you think is abdominal fat?
Line 131: The color was valued, not detected. Did you do blooming before to do color means?
Why do you make three measurements of the pH? At pH45min, pH24h and pH48h?
Results
Line 328: “T”he backfat
Figure 10: What are A and B? Are they a and b?
Discussion
Lines 362-366: references to pig are need.
The antioxidant effect of the proposed diet should be focused on the composition of fatty acids. This study does not. Only the antioxidant influence is related to the color of the meat. This aspect has been insufficiently treated in the document. That is a problem because authors have references “the demands for high-quality pork have grown steadily with the continuously increasing living standards of consumers. Dietary manipulation by antioxidants 377 such as soy isoflavones has been demonstrate to effectively decrease fat and increased 378 lean in growing barrows” (Lines 376-378).
Line 384: The sentence "the carcass weight and dressing percentage by dietary compound antioxidants" is not justified in the text. What is the effect of antioxidants on muscle growth?
The relationship between the microbiota and meat quality results is not clear. The effect of the antioxidant in the ration is not evident in the differences in meat quality between the two treatments.
Conclusion
The microbiota content at the beginning of the experience should have been carried out. The evolution of the microbiota should be checked during the experiment with the two treatments. It may happen that the initial microbiota of the animals is different between individuals and that the antioxidant inclusion factor cannot be verified in the experiment.
Perhaps this is the main problem to support the conclusions of the document.
All the best,
Reviewer 4 Report
My recommendation is that the document can be published in this version.
Author Response
Thank you very much for your time involved in reviewing the manuscript and thanks for your valuable comments and kind recommendation.
Reviewer 5 Report
The article is interesting, well done and may be published in a special issue of "Oxidative Stress in Livestock and Poultry".
Detailed comments:
Line 25 and 178 - provide the correct name of the statistical test.
Line 56 - provide a research hypothesis,
Line 65 - report the approval number and year of the animal testing commission
Table 1: Methionine – pure or DL form? Report the value of the metabolic energy in MJ / kg,
SID - explain abbreviation under Table 1.
Line 94 - enter the number of animals in the pen in brackets
Line 101 and others - it's better to use the term longissimus lumborum
Table 2 - Complete with the name: BW, ADG, ADFI
Table 3 – Better Total cholesterol and please add the values for HDL and LDL cholesterol
Line 227 - better - perirenal fat
Figure 5-8 are interesting, albeit not legible, too small letters, but it will probably remain so.
Line 362 - There have been studies with selenomethionine as a source of selenium - this should be mentioned in the discussion.
Line 500 - give this fragment of the text more precisely.

Round 2
